# Effect of Inulin on Organic Acids and Microstructure of Synbiotic Cheddar-Type Cheese Made from Buffalo Milk

**DOI:** 10.3390/molecules27165137

**Published:** 2022-08-12

**Authors:** Mahad Islam, Maha A. Alharbi, Nada K. Alharbi, Saima Rafiq, Muhammad Shahbaz, Shamas Murtaza, Nighat Raza, Umar Farooq, Muqarrab Ali, Muhammad Imran, Shafaqat Ali

**Affiliations:** 1Department of Food Science & Technology, MNS-University of Agriculture, Multan 60000, Pakistan; 2Department of Biology, College of Science, Princess Nourah bint Abdulrahman University, P.O. Box 84428, Riyadh 11671, Saudi Arabia; 3Department of Food Science and Technology, Faculty of Agriculture, University of Poonch, Rawalakot 12350, Pakistan; 4Department of Agronomy, Muhammad Nawaz Shareef University of Agriculture, Multan 66000, Pakistan; 5Department of Food Science and Technology, University of Narowal, Circular Road, Narowal 51600, Pakistan; 6Food, Nutrition and Lifestyle Unit, King Fahed Medical Research Center, Clinical Biochemistry Department, Faculty of Medicine, King Abdulaziz University, Jeddah 21589, Saudi Arabia; 7Department of Environmental Sciences and Engineering, Government College University, Allama Iqbal Road, Faisalabad 38000, Pakistan; 8Department of Biological Sciences and Technology, China, Medical University, Taichung 40402, Taiwan

**Keywords:** synbiotic cheese, *Bifidobacterium animalis* subsp. *lactis*, inulin, organic acids, probiotics viability, SEM

## Abstract

The current study aimed to produce synbiotic cheese, adding inulin and *Bifidobacterium animalis* subsp. *lactis* as prebiotics and probiotics, respectively. The physicochemical analysis, minerals and organic acids content, sensory evaluation, and probiotic count of the cheese were performed during the ripening. The significant effect of inulin (*p* ≤ 0.01) was found during the ripening period, and changes in physiochemical composition, minerals, and organic acid contents were also observed. Scanning electron microscopy (SEM) of the cheese revealed that inulin could improve the cheese structure. Meanwhile, inulin increased the likeliness of the cheese, and its probiotic viability remained above 10^7^ colony forming unit (CFU) per gram during ripening.

## 1. Introduction

The consciousness of diet and its health-related effects has led toward greater demand for functional food. Functional food carries additional health assistance along with its common nutritional benefits [1]. The decline in customers’ attention towards common products made from milk has directed food processors to produce innovative dairy products with functional properties [2]. Any food product containing probiotic bacteria is an example of functional food.

The term “Probiotics” is derived from a Greek term meaning “for life”. Probiotics are living microbes that offer health assistance to the host when consumed in an appropriate amount [3]. It is known that a decent number of probiotics microorganisms, especially *Bifidobacterium* and *Lactobacillus* genera, need to be absorbed in the intestine to exert health assistance [4]. It is proven that food must hold the minimum level of 10^7^ colony forming unit (cfu) per gram of probiotics at the time of consumption to deliver health assistance [5]. These health benefits include the inhibition and treatment of GI disorders, absorption of vitamins and minerals, the reduction of lactose intolerance, and the inhibition of cardiovascular diseases and some types of cancer [6].

Probiotics may be more effectively delivered to humans by means of fermented milk products. The dairy fermented products comprise various beneficial microbes and are a significant part of our diet. One example of such a dairy fermented product is cheese. Cheese is the protein/casein enriched, developed, or undeveloped fermented dairy product produced by the coagulation of milk. It is the most diversified product among other dairy products with a broad range of flavours and forms, and it is estimated that there are 2000–4000 cheese varieties present globally [7]. Cheese containing probiotics has previously been made successfully by several researchers who carefully selected an appropriate probiotic bacterium to be delivered in the human colon [8,9].

Cheese has been the focus of several markets and research studies in recent years as a suitable option for delivering probiotics into the gut. Cheese offers more benefits as a probiotic carrier as compared to other fermented dairy products, i.e., yoghurt. It acts as a barrier in contradiction of the low pH condition in the gastrointestinal tract (GIT), making it easier for probiotics to survive during gastric transit [10]. Furthermore, the thick matrix and relatively high-fat content of cheeses (i.e., Cheddar) may protect probiotics when reaching the stomach. Another technique that is gaining attention among researchers and dairy product producers is the inclusion of certain food components in meals, which might help bacteria survive longer; prebiotics are an example of such a component.

Prebiotics are small-chain carbohydrates that cannot be digested in humans by gastrointestinal enzymes, and are also acknowledged as tough starch [11]. The non-digestible carbohydrates must accomplish the following criteria to be considered as a prebiotic: (i) prebiotics need to be fermented by probiotics; (ii) prebiotics must have the capability to improve the activity, as well as the viability, of useful bacteria; and (iii) they may survive mammalian enzymes and the gastric acidic environment. It is known that prebiotics can progress the probiotic’s survival into the colon, consequently serving and enhancing the health aids to the host. Due to this reason, prebiotics can be added to probiotic products [12]. Some of the beneficial effects of prebiotics include a reduction of anxiety and depression, a drop in the blood low-density lipoprotein level, increased absorbability of calcium, the incentive of the immunological system, maintenance of accurate intestinal pH value, mitigation of symptoms of vaginal mycosis and peptic ulcers, and low caloric value [13,14].

Synbiotic food is a food mixture comprised of live microbes and substrate(s) which are selectively utilized by the host and provide health benefits [15]. The addition of prebiotic substances in probiotic foods also comes under the category of synbiotic food, and it can raise probiotic viability and stability. Furthermore, it can provide a suitable environment for the growth of beneficial bacteria. The right proportion of pre- and probiotics could result in a perfect synbiotic formulation. Researchers have made several successful attempts for making synbiotic products containing a different combination of pre- and probiotics [16,17,18]. The investigations involved using different types of pre- and probiotics, which were incorporated into different types of products.

The current study was aimed to produce a synbiotic cheese with the addition of *Bifidobacterium animalis* subsp. *lactis* BB-12 (commercial culture) and inulin as a probiotic and prebiotic, respectively, to check its nutritional properties and sensory response. The effect of inulin on the viable count of *B. animalis* was also assessed during the ripening.

## 2. Results and Discussion

### 2.1. Physicochemical Analysis

The chemical composition of cheese is shown in Table 1. The results of ANOVA showed that pH, acidity, moisture, ash, protein, and fat of cheese samples were significantly (*p* ≤ 0.01) influenced by the inulin and ripening period. Indeed, pH, fat, moisture, protein, and ash content of cheese were decreased, while acidity was increased with the increase in inulin concentration.

The results of this study are comparable with the discoveries of Rezaei et al. [19] and Ozturkoglu-Budak et al. [20], who found that with the addition of inulin, pH was decreased, while acidity was increased in synbiotic dairy products. The increase in acidity and decrease in pH may be associated with the conversion of lactose to lactic acid and the production of other organic acids as a natural process. During ripening, a decline in pH and an increase in acidity were observed.

Moisture, protein, ash, and fat contents in synbiotic cheese were found to be lower than in the control treatment. This finding is favored by the conclusion of a study conducted by Barbosa et al. [21], who found a lesser amount of fat and protein contents in synbiotic creamy goat cheese. The lesser concentration of fat and protein in synbiotic cheese was possibly due to the addition of inulin to the cheese mass. Similarly, the moisture and ash content of synbiotic cheese was decreased with the addition of inulin, which is comparable with the findings of Giri et al. [22]. During the ripening period, the moisture percentage of all cheese treatments was decreased, while fat, ash, and protein contents were increased. Hence, the reduced protein and fat content of synbiotic cheese can be overcome during the ripening period. The current findings are also supported by Rafiq et al. [23] and Murtaza et al. [24].

### 2.2. Mineral Profiling

Dairy products have significant mineral content. Among other dairy products, cheese has high minerals, preserving the ability and availability of these minerals throughout the ripening. In this study, three minerals (calcium, sodium, and potassium) were analyzed.

The mean values of minerals of all treatments during the ripening period are shown in Table 2. The results of ANOVA showed a significant effect (*p* ≤ 0.01) of inulin and the ripening period on sodium, potassium, and calcium. Minerals in synbiotic cheese were found to be in a lower concentration, compared to the control cheese. Barbosa et al. [21] also found that cheese containing a probiotic strain and inulin had lesser mineral content. It can be seen in this table that calcium was the major mineral present in all cheese samples, followed by sodium and then potassium. The concentration of sodium is highly dependent upon the NaCl quantity used in cheese manufacturing [25]. During the ripening period, all minerals were found to be increasing.

Regardless of the decrease in mineral contents of cheese by inulin, it was revealed in several studies that inulin can increase the bioavailability of minerals like calcium [18,26]. Hence, the addition of inulin in cheese can have a positive effect on the minerals of the product.

### 2.3. Organic Acids

Organic acids are known to be significant flavour compounds, metabolites, and intermediates of various biochemical processes. The influence of inulin and ripening period on the organic acid’s concentration (citric, lactic, butyric, and acetic) during the ripening of cheese is included in Table 3. A significant (*p* ≤ 0.01) rise in the concentrations of all organic acids was observed during ripening and with inulin concentration. The major organic acids found in all cheese samples were in a proportion: lactic acid (14,245–15,058 ppm) > butyric acid (1862–2002 ppm) > citric acid (828–941 ppm) > acetic acid (421–467 ppm). The same trend was observed in previous studies, and the role of prebiotic was significant in increasing the levels of organic acids [27,28].

The enhanced level of organic acids during the ripening period was also noticed in this study. Research conducted by da Cruz et al. [29] revealed that starter and probiotic bacteria contribute to organic acid production. The increased production of acetic and lactic acids was perhaps due to probiotics activity, in which the acids mentioned above were produced as metabolic products. Moreover, probiotic bacteria can produce extra free amino acids as a result of increased proteolysis, which serves as precursors for the organic acids production at higher concentrations [30].

### 2.4. Microstructure

The microstructure of all cheese treatments with different levels of inulin analyzed by scanning electron microscopy (SEM) when fresh and on the 60th day of ripening is shown in Figure 1a–h. The difference observed in microstructure may be due to the diverse inulin concentration.

The microstructure revealed that all the cheeses containing inulin had a much more uniform and smoother protein matrix. There was no deceptive honeycombing structure observed, which may be because any free water constrained by the inulin can prevent the creation of honeycomb structures. According to Kip et al. [31], if inulin is added during the coagulation and fermentation process, it can become part of the building network of protein by complexion with it. The structure was found to be denser and more compressed in synbiotic cheese, as compared to the probiotic (control) cheese. This finding is similar to the discovery of Sarwar et al. [32].

Over time, as the fat contents started to decrease in the cheese, the size of size globules also decreased. Similarly, the protein matrix started to become denser, with less space between cross-links. This can be seen in Figure 1b,d,f, and h at the end of the ripening period. These outcomes are consistent with the findings of Junyusen et al. [33], who conducted a study on reduced-fat cheese and reported that the reduced fat content improved the establishment of cross-links of the protein matrix, ensuing a denser structural network.

### 2.5. Sensory Evaluation

All the samples of cheese obtained high scores in the sensory analysis. The mean scores of different parameters of sensory evaluation are shown in Figure 2 and Figure 3. The ANOVA showed that the influence of treatment on colour, flavour, and texture was non-significant (*p* > 0.05); however, a highly significant (*p* ≤ 0.01) influence was observed on taste. Overall acceptability was significant (*p* < 0.05). Synbiotic cheese (with inulin) secured good marks as compared to the control treatment (without inulin). All the judges gave scores in the range of 6 (slightly like) to 8 (very much like), on a scale of 1 to 9 (1 being extremely disliked and 9 being extremely liked). The findings of the current study are in accordance with the results of Barbosa et al. [21], who manufactured a creamy goat cheese and found that inulin can increase the likeliness of cheese.

Similarly, an increase in the likeness of cheese was observed with the ripening period. In addition, the flavour of Cheddar cheese was also improved with the ripening time because of the production of flavouring compounds, i.e., organic acids. Hence, the addition of inulin in cheese up to 4% (*w*/*w*) may not have any adverse effect on cheese quality.

### 2.6. Probiotic Count

The viable cell counts of *B. animalis* subsp. *lactis* BB-12 were observed from all treatments during the ripening period, as shown in Figure 4. The statistical analysis (ANOVA) showed that the influence of treatment and ripening on probiotic count was significant (*p* ≤ 0.01). The population of probiotics in all cheese samples was determined to be above 10^6^ to 10^7^ cfu per gram during the ripening period. Synbiotic cheese had a population of 10^7^ cfu per gram, while probiotic cheese (T_0_) had a population of 10^6^ cfu per gram at the end of ripening. It is suggested that food must hold the minimum level of 10^6^–10^7^ cfu per gram of probiotics in order to deliver health assistance. The synbiotic cheeses evaluated in this research had adequate numbers of probiotics over a ripening period to deliver assistance to consumers’ health.

Maximum probiotic count in synbiotic cheese samples was observed on the 20th day of the ripening. The probiotics population was increased due to the availability of inulin as their food during the first 20 days, and after that, it was gradually decreased. A similar pattern was observed by Ozturkoglu-Budak et al. [20]. The main reasons for the loss of probiotic viability are the reduction in pH of the medium and the accumulation of organic acids arising from growth and fermentation. It is concluded that inulin has a synergetic effect on *B. animalis* subsp. *lactis* BB-12 in Cheddar cheese.

## 3. Materials and Methods

### 3.1. Raw Materials Procurement

Buffalos’ milk was collected from a local dairy farm located in the vicinity of Multan, Pakistan. Cheese starter cultures (*Lactococcus lactis* subsp. *lactis* and *Lactococcus lactis* subsp. *cremoris*), probiotic (*B. animalis* subsp. *lactis*; BB-12), and rennet (CHY-MAX^®^ Powder Extra NB) was purchased from Chr. Hansen Ltd. Inulin (Now^®^ foods, Bloomingdale, IL, USA) was procured from a local pharmacy of Multan. All the reagents were made available in the FS&T Department of MNS-University of Agriculture Multan, Pakistan.

### 3.2. Production of Cheese

Four treatments, T_0_ without inulin, T_1_ with 1% (*w*/*w*) inulin, T_2_ with 2% (*w*/*w*) inulin, and T_3_ with 4% (*w*/*w*) inulin of Cheddar cheese were manufactured following the method reported by Murtaza et al. [30]. Only one trial for all the cheese treatments was conducted.

Buffalos’ milk was pasteurized at 63 °C for 20 min. After pasteurization, the temperature of milk was reduced to 31 °C. The cheese starter cultures (*L. lactis* subsp. *lactis* and *L. lactis* subsp. *cremoris*), probiotic (*B. animalis* subsp. *lactis*; BB-12) (in all treatments), and inulin (in T_1_, T_2_, and T_3_) were added to the milk, and gentle stirring was done. After that, milk was left to ripen for 45 min. Then, rennet was added into ripened milk and it was left for 45 min at 30 °C for the coagulation process. After proper curdling, the obtained curd was cut into about 8 mm cubes with the help of a knife. The cooking of curd was done at 38 °C with a gradual increase in temperature of 1 °C in every five minutes with constant stirring until the pH was decreased to 6.2–6.1. Then, the whey was removed from the cheese. After that, the cheddaring process of this whey-drained curd was performed at 38 °C, with constant turning after every 10 min until the pH was decreased to 5.5–5.4. After that, the curd was minced by hand and the salting procedure was done by using 1% (*w*/*w*) NaCl. Salted curd was put into a mold overnight. After pressing, the cheese was removed from the molds and packed into air-tight plastic bags, heat-sealed, and stored at 4–8 °C for further analysis, for 60 days.

### 3.3. Physicochemical Composition

The pH was measured using a pH meter (HI99161, HANNA^®^, Woonsocket, RI, USA) according to the protocol described by Rafiq et al. [23]. Moisture percentage was determined with the help of moisture analyzer (MB120, OHAUS^®^, Parsippany, NJ, USA) [34]. Acidity and ash were determined by following the protocols of AOAC [35]. Fat content was determined by following the protocols of IDF [36], whereas protein content was determined according to the protocol described in IDF [37]. The analysis was p at the intervals of 0, 20, 40 and 60 days during ripening.

### 3.4. Mineral Profile

The determination of minerals (sodium, potassium, and calcium) was carried out using a flame photometer according to the described reported method of Sulieman et al. [38].

Three grams from the cheese samples were ignited at 600 °C overnight, then wetted using distilled water and transferred through a filter paper into a 100 mL conical flask. Then, 10 mL of concentrated HCl was added to the conical flask, and the volume was made up to 100 mL using distilled water. One gram of metal (KCl or NaCl) was weighed and dissolved in one liter of distilled water for stock solution preparation. The standard solution of different concentrations was prepared from this stock solution. Then, readings were recorded using flame photometry and the standard curve was drawn.

### 3.5. Estimation of Organic Acids

The synbiotic cheese was subjected to organic acid profiling through HPLC (high-performance liquid chromatography) according to the method reported by Murtaza et al. [25].

A 7 g sample was mixed in 40 mL of buffer-acetonitrile mobile phase, and extracted for one hour. It was shaken on an agitator (Ultra Turrax, IKA^®^, Staufen, BW, Germany) for 5 min at 4400× *g*, and then centrifugation was performed for five minutes at 6000× *g*. A 0.45-µm membrane filter was used to filter the supernatant twice, which was then directly used for HPLC analysis. A SHIMADZU liquid chromatograph (LC-10 AT VP Series, Shimadzu Corporation, Kyoto, Japan) was equipped with a system regulator injector (SCL-10AVP, Shimadzu Corporation, Kyoto, Japan), fitted with an SPD-10AVP UV-VIS detector (set at 214 nm) (Shimadzu Corporation, Kyoto, Japan) and a sample loop of 20-lL. The conditions of operation were mobile phase, 0.2% (*v*/*v*) 0.049 M acetonitrile adjusted to 2.24 pH with H_3_PO_4_, aqueous 0.5% (*w*/*v*) 0.038 M (NH_4_)_2_HPO_4_, with ambient column temperature and flowing speed of 0.5 mL per minute. A reverse-phase Shim-pack C_18_ (LC) column was used. The reagents used as standards were HPLC-grade (Sigma-Aldrich, St. Louis, MO, USA). Standard solutions and solvents were both filtered with the help of a membrane filter (0.45-µm).

### 3.6. SEM Analysis

Cheeses were examined by using SEM (Vega3, TESCAN, Brno, Czech Republic), by following the method of Jooyandeh et al. [39].

Cheese samples were cut with a razor blade from the interior and immediately fixed in glutaraldehyde in sodium sulfate buffer for two days at 7 °C. The cheeses were dehydrated in graded ethanol series, defatted with the help of chloroform, and then again dehydrated with absolute ethanol. These samples were then mounted on aluminum SEM stubs and examined in the scanning electron microscope.

### 3.7. Sensory Evaluation

The prepared cheeses were evaluated for sensorial characteristics (colour, taste, flavour, texture, and overall acceptability) from a trained panelist of the FS&T Department, MNS-UAM, Multan. Nine-point hedonic scales were used for the rating of the cheese, following the method as used by Shaukat et al. [40].

### 3.8. Survival of Probiotics

The total probiotic population was assessed through the method described by Ozturkoglu-Budak et al. [20] during ripening for comparative studies.

Each sample (10 g) was mixed with Ringer solution (90 mL), followed by homogenizing (2 min), and then subjected to consecutive dilutions. The pour plate technique was used for counting. The population of *B. animalis* subsp. *lactis* BB-12 were enumerated on MRS agar containing NNPL (15 mg/L nalidixic acid, 200 mg/L paromomycin sulfate, 0.5 mg/L l-cysteine chloride, 100 mg/L neomycin sulfate, and 3 mg/L lithium chloride) after incubation for 3 days at 37 °C under an anaerobic environment.

### 3.9. Statistical Analysis

All measurements were done in triplicate. Results drawn from the study were statistically analyzed using the analysis of variance (ANOVA) technique, with a 95% confidence level for the comparison of the findings [41].

## 4. Conclusions

The current study showed that inulin and *B. animalis* subsp. *lactis* BB-12 can be successfully added to cheese to make a synbiotic product that can increase cheese’s functional properties. The nutritional profile of synbiotic cheese containing up to 4% (*w*/*w*) inulin is almost similar to the normal cheese, as all the values are within the range of cheddar type cheese according to the literature. The organic acid contents—which act as flavouring compounds—increased, and the mineral content was decreased with the addition of inulin; however, an increase in the contents as mentioned above was observed during the ripening period. Additionally, it was revealed by the SEM images that inulin can improve the structure of synbiotic cheese. It was also found that probiotics can successfully be delivered in Cheddar-type cheese, and their viability will remain above 10^7^ cfu per gram till 60 days of storage with inulin addition. Although the probiotic count in the control cheese also remained above 10^7^ cfu per gram, the decline pattern of probiotics shows that this viability will reduce sooner than in the synbiotic cheese. To check the viability of probiotics for a longer period of time, study can be extended for more than 60 days. The recipe of synbiotic cheese received great appraisal by the judges because of its better sensorial characteristics than the control cheese. The synbiotic Cheddar-type cheeses produced in this research may be potential carriers of pre- and probiotics, and may bring many health benefits to our bodies.

## Figures and Tables

**Figure 1 molecules-27-05137-f001:**
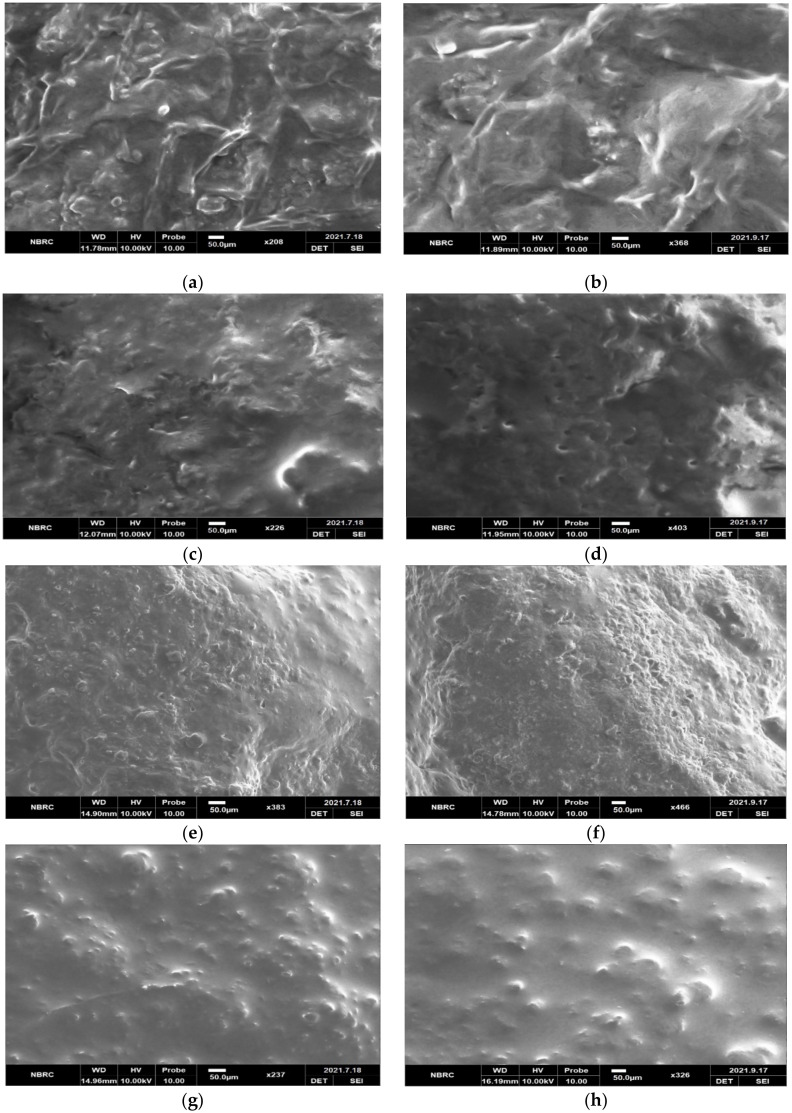
Scanning electron microscopy (SEM) images of cheese: (**a**) T_0_ at 0 day; (**b**) T_0_ at 60th day; (**c**) T_1_ at 0 day; (**d**) T_1_ at 60th day; (**e**) T_2_ at 0 day; (**f**) T_2_ at 60th day; (**g**) T_3_ at 0 day; (**h**) T_3_ at 60th day.

**Figure 2 molecules-27-05137-f002:**
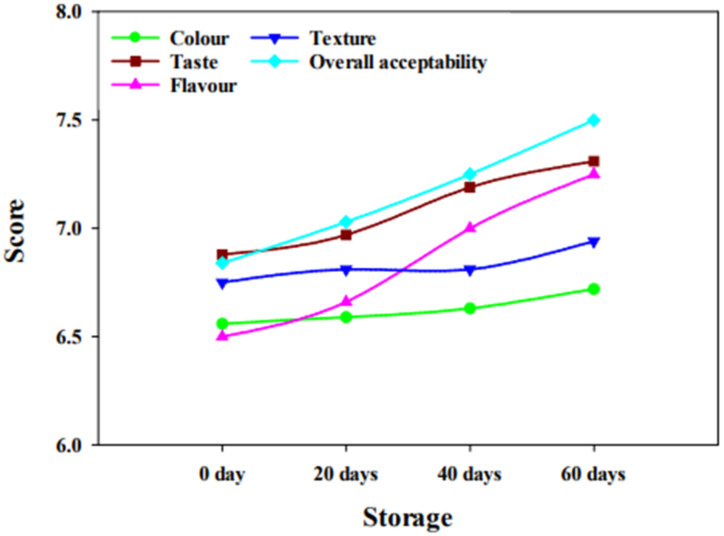
Effect of ripening on sensory properties of cheese.

**Figure 3 molecules-27-05137-f003:**
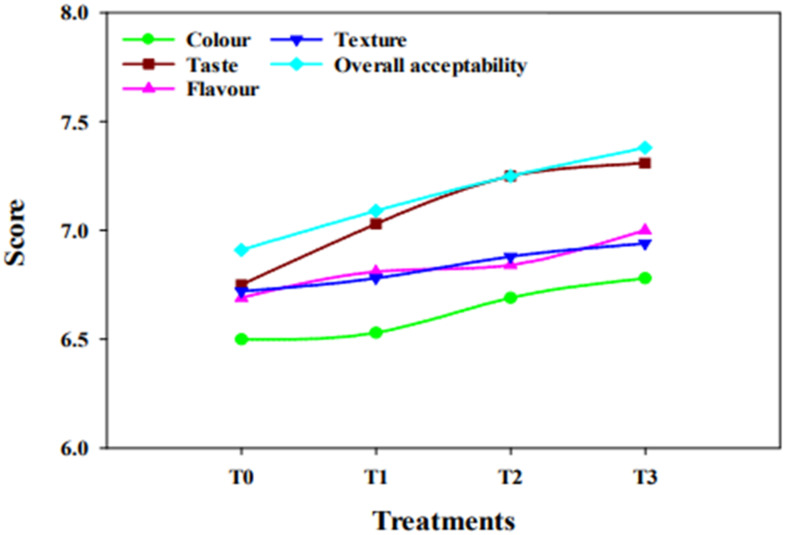
Effect of treatment on sensory properties of cheese.

**Figure 4 molecules-27-05137-f004:**
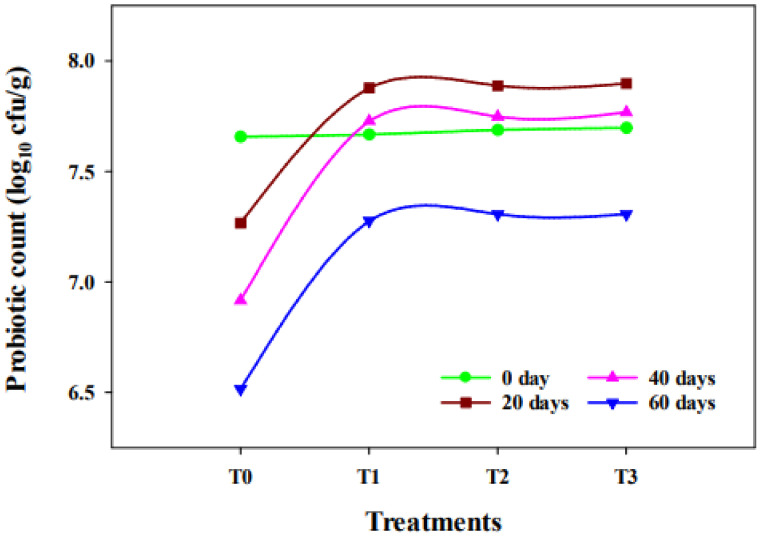
Effect of treatment and ripening on viable probiotic count.

**Table 1 molecules-27-05137-t001:** Composition of all cheese treatments during the ripening period (means ± standard deviation, SD).

Parameter	Treatments	Ripening				Means
		0 Day	20 Days	40 Days	60 Days	
Moisture %	T_0_	38.67 ± 0.02 ^a^	38.36 ± 0.01 ^b^	38.07 ± 0.01 ^d^	37.78 ± 0.01 ^g^	38.22 ± 0.01 ^a^
	T_1_	38.23 ± 0.01 ^c^	37.97 ± 0.02 ^f^	37.74 ± 0.01 ^gh^	37.50 ± 0.02 ^ij^	37.86 ± 0.01 ^b^
	T_2_	38.03 ± 0.01 ^e^	37.73 ± 0.01 ^h^	37.48 ± 0.02 ^j^	37.22 ± 0.01 ^l^	37.61 ± 0.01 ^c^
	T_3_	37.77 ± 0.02 ^g^	37.53 ± 0.01 ^i^	37.36 ± 0.01 ^k^	37.11 ± 0.02 ^m^	37.44 ± 0.01 ^d^
Fat %	T_0_	31.32 ± 0.02 ^d^	31.43 ± 0.03 ^c^	31.52 ± 0.03 ^b^	31.63 ± 0.03 ^a^	31.48 ± 0.02 ^a^
	T_1_	31.13 ± 0.03 ^f^	31.22 ± 0.03 ^e^	31.33 ± 0.03 ^d^	31.42 ± 0.03 ^c^	31.28 ± 0.03 ^b^
	T_2_	31.01 ± 0.02 ^g^	31.11 ± 0.02 ^f^	31.22 ± 0.03 ^e^	31.33 ± 0.03 ^d^	31.17 ± 0.02 ^c^
	T_3_	30.88 ± 0.03 ^h^	31.01 ± 0.02 ^g^	31.11 ± 0.02 ^f^	31.22 ± 0.03 ^e^	31.06 ± 0.02 ^d^
Protein %	T_0_	27.11 ± 0.01 ^f^	27.28 ± 0.02 ^d^	27.43 ± 0.01 ^b^	27.59 ± 0.02 ^a^	27.35 ± 0.02 ^a^
	T_1_	26.94 ± 0.01 ^h^	27.12 ± 0.01 ^f^	27.28 ± 0.01 ^d^	27.42 ± 0.02 ^b^	27.19 ± 0.01 ^b^
	T_2_	26.85 ± 0.01 ^i^	27.02 ± 0.01 ^fg^	27.20 ± 0.02 ^e^	27.33 ± 0.01 ^c^	27.10 ± 0.01 ^c^
	T_3_	26.72 ± 0.01 ^j^	26.91 ± 0.01 ^h^	27.06 ± 0.01 ^g^	27.21 ± 0.01 ^e^	26.97 ± 0.01 ^d^
Ash %	T_0_	3.956 ± 0.002 ^i^	3.977 ± 0.002 ^gh^	3.998 ± 0.002 ^d^	4.027 ± 0.002 ^a^	3.989 ± 0.002 ^a^
	T_1_	3.949 ± 0.002 ^j^	3.971 ± 0.001 ^h^	3.990 ± 0.001 ^e^	4.020 ± 0.002 ^ab^	3.982 ± 0.001 ^b^
	T_2_	3.941 ± 0.001 ^k^	3.962 ± 0.002 ^i^	3.985 ± 0.001 ^ef^	4.014 ± 0.001 ^b^	3.976 ± 0.001 ^c^
	T_3_	3.932 ± 0.002 ^l^	3.943 ± 0.001 ^jk^	3.980 ± 0.002 ^fg^	4.006 ± 0.002 ^c^	3.964 ± 0.002 ^d^
pH	T_0_	5.35 ± 0.01 ^a^	5.28 ± 0.01 ^c^	5.21 ± 0.01 ^e^	5.12 ± 0.01 ^g^	5.24 ± 0.01 ^a^
	T_1_	5.32 ± 0.01 ^b^	5.25 ± 0.01 ^d^	5.18 ± 0.01 ^f^	5.09 ± 0.01 ^h^	5.21 ± 0.01 ^b^
	T_2_	5.30 ± 0.01 ^c^	5.22 ± 0.01 ^d^	5.16 ± 0.01 ^f^	5.07 ± 0.01 ^i^	5.19 ± 0.01 ^c^
	T_3_	5.29 ± 0.01 ^c^	5.21 ± 0.01 ^e^	5.14 ± 0.01 ^g^	5.05 ± 0.01 ^i^	5.17 ± 0.01 ^d^
Acidity %	T_0_	0.898 ± 0.001 ^h^	0.902 ± 0.001 ^fg^	0.906 ± 0.001 ^de^	0.911 ± 0.001 ^bc^	0.904 ± 0.001 ^d^
	T_1_	0.900 ± 0.001 ^gh^	0.904 ± 0.001 ^fg^	0.908 ± 0.001 ^cd^	0.912 ± 0.001 ^abc^	0.906 ± 0.001 ^c^
	T_2_	0.902 ± 0.001 ^fg^	0.905 ± 0.001 ^ef^	0.910 ± 0.001 ^abc^	0.913 ± 0.001 ^ab^	0.908 ± 0.001 ^b^
	T_3_	0.905 ± 0.001 ^ef^	0.907 ± 0.001 ^de^	0.913 ± 0.001 ^ab^	0.915 ± 0.001 ^a^	0.910 ± 0.001 ^a^

T_0_ = Control cheese; T_1_ = Synbiotic cheese with 1% (*w*/*w*) inulin; T_2_ = Synbiotic cheese with 2% (*w*/*w*) inulin; T_3_ = Synbiotic cheese with 4% (*w*/*w*) inulin. The mean values with dissimilar letters (a–l) inside the table have statistically significant differences (*p* ≤ 0.01).

**Table 2 molecules-27-05137-t002:** Mineral profile of all cheese treatments during the ripening period (means ± SD).

Minerals	Treatments	Ripening				Means
		0 Day	20 Days	40 Days	60 Days	
Sodium (mg)	T_0_	680.46 ± 0.01 ^j^	680.98 ± 0.01 ^f^	681.48 ± 0.02 ^c^	681.98 ± 0.01 ^a^	681.22 ± 0.01 ^a^
	T_1_	680.11 ± 0.02 ^m^	680.55 ± 0.01 ^i^	681.10 ± 0.01 ^e^	681.57 ± 0.01 ^b^	680.83 ± 0.01 ^b^
	T_2_	679.76 ± 0.01 ^o^	680.21 ± 0.01 ^l^	680.75 ± 0.01 ^h^	681.26 ± 0.01 ^d^	680.50 ± 0.01 ^c^
	T_3_	679.40 ± 0.01 ^p^	679.88 ± 0.01 ^n^	680.39 ± 0.02 ^k^	680.88 ± 0.01 ^g^	680.14 ± 0.01 ^d^
Potassium (mg)	T_0_	81.40 ± 0.01 ^k^	81.66 ± 0.02 ^g^	81.93 ± 0.01 ^c^	82.18 ± 0.01 ^a^	81.79 ± 0.01 ^a^
	T_1_	81.27 ± 0.01 ^m^	81.53 ± 0.01 ^i^	81.78 ± 0.01 ^e^	82.05 ± 0.01 ^b^	81.66 ± 0.01 ^b^
	T_2_	81.10 ± 0.01 ^n^	81.36 ± 0.02 ^l^	81.63 ± 0.01 ^h^	81.89 ± 0.01 ^d^	81.50 ± 0.01 ^c^
	T_3_	80.98 ± 0.02 ^o^	81.24 ± 0.01 ^m^	81.49 ± 0.01 ^j^	81.73 ± 0.01 ^f^	81.36 ± 0.01 ^d^
Calcium (mg)	T_0_	749.23 ± 0.01 ^i^	749.55 ± 0.01 ^f^	749.88 ± 0.01 ^c^	750.20 ± 0.01 ^a^	749.72 ± 0.01 ^a^
	T_1_	749.11 ± 0.01 ^j^	749.43 ± 0.02 ^g^	749.74 ± 0.01 ^d^	750.07 ± 0.01 ^b^	749.59 ± 0.01 ^b^
	T_2_	748.96 ± 0.01 ^k^	749.30 ± 0.01 ^h^	749.45 ± 0.01 ^e^	749.92 ± 0.01 ^c^	749.44 ± 0.01 ^c^
	T_3_	748.81 ± 0.01 ^l^	749.15 ± 0.01 ^j^	749.46 ± 0.02 ^g^	749.77 ± 0.02 ^d^	749.29 ± 0.01 ^d^

T_0_ = Control cheese; T_1_ = Synbiotic cheese with 1% (*w*/*w*) inulin; T_2_ = Synbiotic cheese with 2% (*w*/*w*) inulin; T_3_ = Synbiotic cheese with 4% (*w*/*w*) inulin. The mean values with dissimilar letters (a–p) inside the table have statistically significant differences (*p* ≤ 0.01).

**Table 3 molecules-27-05137-t003:** Organic acids of all cheese treatments during the ripening period (means ± SD).

Organic Acids	Treatments	Ripening				Means
		0 Day	20 Days	40 Days	60 Days	
Lactic acid (ppm)	T_0_	14,245 ± 3 ^o^	14,476 ± 3 ^k^	14,753 ± 4 ^g^	14,902 ± 5 ^d^	14,594 ± 4 ^d^
	T_1_	14,298 ± 6 ^n^	14,525 ± 4 ^j^	14,803 ± 5 ^f^	14,951 ± 4 ^c^	14,644 ± 5 ^c^
	T_2_	14,351 ± 6 ^m^	14,588 ± 5 ^i^	14,852 ± 4 ^e^	15,005 ± 5 ^b^	14,699 ± 5 ^b^
	T_3_	14,409 ± 5 ^l^	14,652 ± 5 ^h^	14,911 ± 5 ^d^	15,058 ± 6 ^a^	14,758 ± 5 ^a^
Acetic acid (ppm)	T_0_	421.0 ± 1.0 ^i^	428.0 ± 1.0 ^h^	435.7 ± 0.6 ^f^	442.7 ± 0.6 ^f^	431.8 ± 0.8 ^d^
	T_1_	425.7 ± 0.6 ^h^	435.0 ± 1.0 ^f^	443.3 ± 0.6 ^e^	450.7 ± 0.6 ^c^	438.6 ± 0.7 ^c^
	T_2_	430.7 ± 0.6 ^g^	441.3 ± 0.6 ^e^	450.0 ± 1.0 ^c^	458.3 ± 0.6 ^b^	445.1 ± 0.7 ^b^
	T_3_	436.3 ± 0.6 ^f^	447.0 ± 1.0 ^d^	457.3 ± 1.1 ^b^	467.7 ± 1.1 ^a^	452.0 ± 1.0 ^a^
Citric acid (ppm)	T_0_	828.3 ± 1.1 ^l^	849.7 ± 1.1 ^j^	871.7 ± 1.5 ^g^	896.7 ± 1.5 ^d^	861.6 ± 1.3 ^d^
	T_1_	841.3 ± 0.6 ^k^	863.7 ± 0.6 ^h^	886.0 ± 1.0 ^e^	912.0 ± 1.0 ^c^	875.7 ± 0.8 ^c^
	T_2_	853.6 ± 0.6 ^i^	875.7 ± 0.6 ^f^	899.7 ± 0.6 ^d^	926.3 ± 1.1 ^b^	888.8 ± 0.7 ^b^
	T_3_	862.7 ± 1.1 ^h^	887.0 ± 1.1 ^e^	913.7 ± 0.6 ^c^	941.0 ± 1.0 ^a^	901.1 ± 0.9 ^a^
Butyric acid (ppm)	T_0_	1862.0 ± 2.0 ^m^	1896.3 ± 2.1 ^j^	1931.0 ± 1.0 ^g^	1968.0 ± 1.0 ^d^	1914.3 ± 1.5 ^d^
	T_1_	1874.3 ± 1.5 ^l^	1908.0 ± 1.0 ^i^	1943.7 ± 1.5 ^f^	1980.0 ± 1.7 ^c^	1926.5 ± 1.4 ^c^
	T_2_	1886.3 ± 1.5 ^k^	1919.7 ± 1.5 ^h^	1955.0 ± 1.0 ^e^	1992.0 ± 1.0 ^b^	1938.3 ± 1.3 ^b^
	T_3_	1899.7 ± 1.5 ^j^	1930.7 ± 1.2 ^g^	1966.3 ± 1.5 ^d^	2002.3 ± 0.6 ^a^	1949.8 ± 1.2 ^a^

T_0_ = Control cheese; T_1_ = Synbiotic cheese with 1% (*w*/*w*) inulin; T_2_ = Synbiotic cheese with 2% (*w*/*w*) inulin; T_3_ = Synbiotic cheese with 4% (*w*/*w*) inulin. The mean values with dissimilar letters (a–o) inside the table have statistically significant differences (*p* ≤ 0.01).

## Data Availability

Not applicable.

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
