# Peer review of "Effect of Inulin on Organic Acids and Microstructure of Synbiotic Cheddar-Type Cheese Made from Buffalo Milk"

_molecules, 2022, doi:10.3390/molecules27165137_

Round 1

Reviewer 1 Report

Dear Editor and Authors,

I send you my review about the article “Effect of inulin on organic acids and microstructure of synbiotic cheddar-type cheese”.

The scope of the paper, as reported in the aim was to produce synbiotic cheese, adding inulin and Bifidobacterium animalis as prebiotics and probiotics, respectively.

In my opinion, the paper, although it result sufficiently well written and it is sufficiently well structured, however, it show some lack and, therefore, my opinion is that it result suitable for publication after some minor revisions that I a report below.

The introduction result adequately to the aim of the research. However, in the introduction should be better explained the originality of this paper, that in the present version, result a little lacking. To improve the originality of the paper, I suggest to the Authors, to reports the recents article that have study the same aspects of their paper. After, they should stress, always in the introduction, the difference among their study and the others previously reported.

The results is well presented and they are well discussed, also in comparison to the data reported in the literature.

However, to facilitate the reading of the graphs by the readers and for a better understanding of the data shown in the figures 2, 3 and 4 it should be highlighted, in the graphs the difference among the means values, for examples using different letters.

Moreover, always to a better understanding of the data by the readers It should be better to report in the tables or in the caption of the tables the significance of the difference among the means.

The paragraph materials and methods needs to be improved adding some information.

Indeed, in the materials and methods chapter should be reported the number of the cheese-making trials made and the sampling procedure of the cheese samples.

Furthermore, in my opinion, the difference among the number of the cheese-making trials and the number of replicate of analysis should be better highlighted.

Moreover, always in materials and methods the model used to estimated the least square means.

Finally, the conclusion resulted adequate to the data showed and to the aim of the research.

Best regards

Author Response

Greetings!

thanks for your kind words and comments.

the amendments have been made. kindly find the attached file.

Regards

Reviewer 2 Report

Article is interesting but I suggest to check results and statistical analysis. You showed statistical difference but Standard deviation is too small. I doubt how did you repeat experiment and analysis as neccesery for statistics. My comments were given in pdf text

Author Response

Greetings,

thanks for reviewing this manuscript. I have tried to improve this manuscript by keeping in mind your kind comments. 

Regards

Round 2

Reviewer 2 Report

Authors did review of article according to my suggestion and think its improved.